# Exploring the association between sleep quality, internet addiction, and related factors among adolescents in Dakshinkali Municipality, Nepal

Sophiya Acharya[1]☯, Anisha Chalise[2], Nirmal Raj Marasine[ID][3], Shishir Paudel[ID][1,4]☯ *

1 Department of Public Health, CiST College, Pokhara University, Kathmandu, Nepal, 2 Center for Research on Environment, Health and Population Activities (CREHPA), Lalitpur Nepal, 3 Department of Pharmacy, CiST College, Pokhara University, Kathmandu, Nepal, 4 Kathmandu Institute of Child Health, Hepaliheight, Kathmandu, Nepal

☯ These authors contributed equally to this work.
* shishirpaudel11@gmail.com

## Abstract

### Background

Poor sleep quality and internet addiction are significant issues affecting adolescents globally, and Nepal is no exception. Several studies have independently assessed the prevalence and associated factors of poor sleep quality and internet addiction among Nepali adolescents and youth, but the relationship between sleep-related attributes and internet addiction remains unexplored. This study aimed to explore the prevalence and contributing factors of poor sleep quality and internet addiction along with the relationship between sleep quality-related attributes and internet addiction.

### Material and methods

A cross-sectional study was conducted among 243 adolescents of Dakshinkali Municipality, Nepal. Pittsburgh Sleep Quality Index and Young's Internet Addiction Test scale were used to measure sleep quality and internet addiction. Pearson's chi-square test and binary logistic regression were performed at a 5% level of significance to examine the associated factors.

### Results

The prevalence of poor sleep quality was 27.6% (95% CI: 22.6–33.7) while potential internet addiction was 49.4% (95% CI: 42.0–56.7). Poor sleep quality was associated with internet addiction (aOR: 1.845; 95% CI: 1.344–3.608), poor perceived relation with teachers (aOR: 2.274; 95% CI: 1.149–4.497), and presence of family conflict (aOR: 2.355; 95% CI: 1.040–5.329). Bad subjective sleep quality (aOR: 5.613; 95% CI: 2.007–15.701), sleep disturbance (aOR: 1.781; 95% CI: 1.251–4.872), frequent daytime dysfunction (aOR: 1.902; 95% CI: 1.083–4.638), and poor perceived relation with teachers (aOR: 2.298; 95% CI: 1.233–

**Data Availability Statement:** All relevant data are within the manuscript and its Supporting Information files.

**Funding:** The author(s) received no specific funding for this work.

**Competing interests:** The authors have declared that no competing interests exist.

4.285), and presence of family conflict (aOR: 1.606; 95% CI: 1.202–3.675) were associated with internet addiction.

## Conclusion

Almost a quarter of adolescents' experience poor sleep quality, while nearly half screened positive for potential internet addiction. Established interrelations between sleep quality and internet usage underscore the importance of integrated intervention approaches combining lifestyle modification and family/school support to protect and promote the mental health and well-being of Nepalese adolescents.

## Introduction

Sleep quality encompasses a multi-faceted concept comprising different sleep attributes such as sleep duration, efficiency, latency, disturbances, daytime somnolence, and individual's subjective sleep experience [1]. Adequate and quality sleep is vital for physical and mental well-being, especially during adolescence, where it plays a crucial role in normal physical development as sleep duration and overall sleep quality, could also influence growth hormone secretion [2]. For optimum health and performance, adolescents are recommended to get 8 to 10 hours of sleep daily [3]. However, many adolescents fail to get quality sleep, resulting in an experience of unfavorable physiological, mental, and social stress [3,4].

The digital age has transformed communication, information exchange, and connectivity. While the internet offers undeniable benefits, concerns have emerged regarding its potential impact on individual well-being, particularly among adolescents [5,6]. Internet addiction, a condition characterized by uncontrolled and excessive internet use to the detriment of daily life, has become a growing concern due to its negative influence on social skills and mental health, mostly among adolescents [7–11]. Excessive internet use may disrupt sleep patterns, making it difficult to fall asleep or stay asleep, leading to sleep deprivation and its associated health problems [12]. Adolescents who don't get enough sleep may be more prone to excessive internet use, seeking stimulation or distraction from their sleepiness, which further exacerbates sleep difficulties.

The higher prevalence of poor sleep quality has been noted among adolescents in different nations throughout the world [13–17]. A meta-analysis of 20 studies from 12 countries found 37% of adolescents experience sleep problems [18]. In Nepal, multiple studies have documented concerningly high levels of sleep disturbance among students at various academic levels, with the prevalence of poor sleep quality ranging from 31% to 60.9% [7,19–22]. Internet addiction follows a similar global pattern. A meta-analysis, encompassing 43 studies from 17 countries, reported that 6.0% of adolescents experience internet addiction [23]. Nepal has also observed an alarming rate of internet addiction or possible internet addiction among adolescents in urban settlements, which ranged between 13.3% and 34.35% [7,24,25]. There is a growing concern about internet addiction and poor sleep quality among adolescents globally, and Nepal is no exception. Several studies have independently assessed the prevalence and associated factors of poor sleep quality and internet addiction among Nepali adolescents and youth, but the relationship between sleep-related attributes and internet addiction remains unexplored. Thus, this study aims to explore the status of and contributing factors for both poor sleep quality and internet addiction among school-going adolescents of Dakshinkali

Municipality. Additionally, it also investigates the association of sleep-related attributes such as duration, latency, and efficiency with internet addiction.

## Materials and methods

### Study design and setting

This cross-sectional study was executed among school-going adolescents of Dakshinkali Municipality of Kathmandu district, Bagmati Province from August to November 2023. Dakshinkali municipality lies in the southern part of Kathmandu district, with 6,489 households housing 26,272 residents, where adolescents aged 10–19 years' account for 16% of the total population of the municipality [26].

### Sample size and sampling procedure

The sample size was determined using Cochran's formula for the estimation of a proportion ($n = z^2 pq/d^2$) at 95% confidence interval and 5% margin of error. A cross-sectional study from western Nepal reported the prevalence of poor sleep quality among school adolescents to be 39.1% [19]. Similarly, another study based on adolescents in the Kathmandu district reported potential internet addiction at 35% [24]. Considering these two proportions, the initial sample size for sleep quality and internet addiction was estimated to be 215 and 209 respectively. To take the optimal sample, we chose an initial sample estimate of 215 and optimized it further to 280 after adjusting 30% non-response rate.

A total of 16 secondary-level schools were functional inside the Dakshinkali Municipality [26]. Out of these, eight schools were selected randomly using Decision Analyst Stat 2.0 software covering 50% of the total schools inside the Municipality. The sample frame was prepared from the school attendance record. The required number of samples to be drawn from each selected school was determined proportionally based on the total number of students enrolled at each school in grades nine and ten. The ninth and tenth-grade classes and sections of each school were coded and out of them, ten classes were randomly selected using the lottery method. All the students present in the selected classes were involved in the study after obtaining informed consent from them. Informed consent was obtained from all the students prior to their participation in the study. The students who failed to provide parental consent and/or their informed consent were excluded from the study.

### Data collection

The data collection was performed in the classroom using a self-administered questionnaire, after acquiring permission from each selected academic institution. The teachers helped to arrange a data collection session for around an hour, where the questionnaire was distributed and the students were oriented. The participants completed the questionnaire in the classroom setup where they were provided with enough space to maintain their privacy. The self-administered questionnaire consisted of four sections where the first section consisted of general information about students' socio-demographic profiles. The second section consisted of the Young's Internet Addiction Test (IAT) [27], aimed to assess the level of internet use and addiction. The third section consisted of questions regarding psychological factors which included the occurrence of stressful life events within the past month, the presence of family conflicts within the past month, and self-rated perceived relation with friends, teachers, and parents. The fourth section consisted of the Pittsburgh Sleep Quality Index (PSQI) [28], to assess the status of sleep quality among the participants. The questionnaire was translated into the Nepali language and then back-translated in English, maintaining translation validity. A native Nepali

speaker with experience in psychometric tool translation performed the initial translation of PSQI and IAT from English to Nepali. A different member of research team, fluent in both English and Nepali and blind to the original English versions, then back-translated the Nepali versions into English. There were minor discrepancies which were resolved by discussion. Pre-test of the tool was performed among 10% of the total sample population (n = 30) in one of the non-sampled schools of Dakshinkali Municipality. The internal consistency of the tools was assessed through the calculation of Cronbach's alpha, which was 0.70 for PSQI and 0.81 for IAT in this study.

## Measurements

This study employed two self-report questionnaires to assess the dependent variables sleep quality and internet addiction among participants. Young's Internet Addiction Test (IAT), is a questionnaire developed by Dr. Kimberly Young [27], which is a 20-item, five-point Likert scale used to measure the severity of self-reported compulsive use of the internet. Total IAT score ranges between 0–100 where a score ≤ 30 represents normal level internet use, between 31–49 represents mild internet addiction, between 50–79 represents moderate internet addiction, and a score >80 represents severe internet dependency [27]. The same original cutoff was used in this study to assess internet addiction among participants. Furthermore, score of ≤ 30 represents normal level internet use while score >30 represents possible/internet addiction. Similarly, the Pittsburgh Sleep Quality Index (PSQI) [28], a 19-item self-reporting questionnaire, was used to evaluate sleep quality over a period of one month, by examining a variety of sleep-related factors. PSQI is a four-point Likert scale ranging from 0 to 3 for each 19 items with the overall PSQI score ranging from 0 to 21. The lower score represents better sleep quality and the cut-off value of ≤5 can be used to denote good sleep quality. Furthermore, the components of sleep quality, such as subjective sleep quality, sleep duration, sleep efficiency, sleep disturbance, use of sleep medications and day time dysfunction were derived from PSQI guideline [28].

The independent variables included socio-demographic characteristics such as, age, gender, family economic status, psychological factors such as stressful life events, family conflict, perceived relationships with parents, teachers, and friends, and specific sleep attributes such as sleep latency, subjective sleep quality, sleep disturbances, and daytime dysfunction.

## Statistical analysis

The data were entered using EpiData (V.3.1) and exported to Statistical Package for the Social Sciences (SPSS V.26) for statistical analysis. Descriptive statistics such as mean, frequency, and percentage were calculated. Pearson chi-square test and binary logistic regression were applied to assess the association between different independent and dependent variables at a 95% confidence interval and 5% level of significance. The variables found to be statistically significant in bivariate analysis were included in the final model for multivariable analysis to calculate the adjusted odds ratio (aOR). The variance inflation factor (VIF) test was performed among selected independent variables to be subjective to the regression model to manage the issue of multicollinearity. The VIF greater than five was taken as an indication of multicollinearity.

## Ethical statement

This study involves human participants and was approved by Institutional Review Committee (IRC) of CiST College (Ref no. 22/080/081). Written informed consent was obtained from all the participants aged 18 and above. Written informed consent from the parents along with written informed consent (assent) from participants was obtained for participants aged 17

years and below. The students below the age of 18 years were provided with parental consent form few days prior to data collection. Those students below the age of 18 who didn't provide signed parental consent on day of data collection were excluded. Confidentiality of participants' data was ensured throughout the study. Participants were informed about their voluntary participation in the study, and were clarified that they had the right to withdraw from the study at any point without facing any repercussions. Permission for conducting research was obtained from respective academic institutions and local governmental authorities prior to data collection.

## Results

A total of 280 students were provided with the questionnaire, where only 243 provided their complete responses to each question, achieving a response rate of 87%. Thus, the result is based on the acquired 243 complete responses. Among the total participants, 67 of them reported having poor sleep quality as per the PSQI cut-off point of >5, illustrating the prevalence of poor sleep quality at 27.6% (95% CI: 22.6–33.7). Similarly, the prevalence of possible/internet addiction was observed to be 49.4% (95% CI: 42.0–56.7) as per Young's Internet Addiction Test cut-off point of >30 (Table 1).

The age of the participants ranges from 12 to 17 years with a mean age of 14.88±0.95 years. Nearly half of the study participants (54.3%) were female. One-fifth (19.8%) of the participants reported having experienced stressful life events within the past month, such as loss of family members or loved ones, parents' divorce, or other stressful events. Majority of participants (79%) reported perceiving a good relationship with their parents while nearly half (59.3%) perceived a good relationship with their friends. Almost one-fifth (18.5%) acknowledged having conflict within their family, such as family misunderstandings, quarrels, and relationship conflicts among parents. From the chi-square test, it was observed that economic status of the family and psychological factors such as stressful life events, family conflict, and perceived relationship with parents and teachers had a statistically significant relationship with sleep quality. Similarly, a statistically significant relationship was observed between internet addiction and participants' experience of stressful life events, family conflict, and their perceived relationship with their friends and teachers (Table 2).

The distribution of Internet addiction across different attributes of sleep quality was compared. A statistically significant relation was observed between internet addiction and overall sleep quality. A higher proportion of poor overall sleep quality (64.2%) was reported among those having internet addiction as compared to normal internet users. It was also observed that internet addiction was associated with sleep quality-related attributes such as sleep latency, subjective sleep quality, increased sleep disturbance, and daytime dysfunction. However.no

**Table 1. Prevalence of poor sleep quality and internet addition.**

| Variables | Frequency (n) | Percentage (95% CI) |
|---|---|---|
| **Sleep Quality** | | |
| Good sleep quality | 176 | 72.4 (66.3–77.4) |
| Poor sleep quality | 67 | 27.6 (22.6–33.7) |
| **Internet Addiction** | | |
| Normal internet use | 123 | 50.6 (43.3–58.0) |
| Mild internet addiction | 83 | 34.2 (28.0–39.9) |
| Moderate internet addiction | 37 | 15.2 (10.7–20.5) |

**Table 2. Socio-demographic and personal factors, and their association with sleep quality and internet addiction.**

| Variables | n (%) | Sleep Quality | | X² (p-value) | Internet Addiction | | X² (p-value) |
|---|---|---|---|---|---|---|---|
| | | Good n (%) | Poor n (%) | | Normal internet use n (%) | Internet addiction n (%) | |
| **Age** | | | | | | | |
| <15 years | 82 (33.7) | 63 (76.8) | 19 (23.2) | 1.201 (0.273) | 40 (48.8) | 42 (51.2) | 0.167 (0.683) |
| ≥15 years | 161 (66.3) | 113 (70.2) | 48 (29.8) | | 83 (51.6) | 78 (48.4) | |
| **Gender** | | | | | | | |
| Male | 111 (45.7) | 86 (77.4) | 25 (22.5) | 2.609 (0.106) | 51 (45.9) | 60 (54.1) | 1.784 (0.182) |
| Female | 132 (54.3) | 90 (68.2) | 42 (31.8) | | 72 (54.5) | 60 (45.5) | |
| **Father's Education** | | | | | | | |
| Illiterate | 30 (12.3) | 17 (56.7) | 13 (43.3) | 4.455 (0.108) | 19 (63.3) | 11 (38.7) | 2.311 (0.315) |
| Non-formal education | 24 (9.9) | 17(70.8) | 7 (29.2) | | 11 (45.8) | 13 (54.2) | |
| Formal education | 189 | 142 (75.1) | 47 (24.9) | | 93 (49.2) | 96 (50.8) | |
| **Mother's Education** | | | | | | | |
| Illiterate | 58 (23.9) | 42 (72.4) | 16 (27.6) | 5.130 (0.077) | 31 (53.4) | 27 (46.6) | 2.339 (0.311) |
| Non-formal education | 29 (11.9) | 16 (55.2) | 13 (44.8) | | 18 (62.1) | 11 (37.9) | |
| Formal education | 156 | 118 (75.6) | 38 (24.4) | | 74 (47.4) | 82 (52.6) | |
| **Family Economic Status** | | | | | | | |
| First quintile | 46 (18.9) | 32 (69.6) | 14 (30.4) | 10.777 (0.029)* | 30 (65.2) | 16 (34.8) | 7.648 (0.105) |
| Second quintile | 51 (21.0) | 32 (62.7) | 19 (37.3) | | 25 (49.0) | 26 (51.0) | |
| Third quintile | 49 (20.2) | 31 (63.3) | 18 (36.7) | | 20 (40.8) | 29 (59.2) | |
| Fourth quintile | 49 (20.2) | 40 (81.6) | 9 (18.4) | | 21 (42.9) | 28 (57.1) | |
| Fifth quintile | 48 (19.8) | 41 (85.4) | 7 (14.6) | | 27 (56.3) | 21 (43.8) | |
| **Stressful Life Events** | | | | | | | |
| Yes | 48 (19.8) | 29 (60.4) | 19 (39.6) | 4.321 (0.038)* | 16 (33.3) | 32 (66.7) | 7.149 (0.008)* |
| No | 195 (80.2) | 147 (75.4) | 48 (24.6) | | 107 (54.9) | 88 (45.1) | |
| **Family Conflict** | | | | | | | |
| Presence | 45 (18.5) | 23 (51.1) | 22 (48.9) | 12.567 (<0.001)** | 16 (35.6) | 29 (64.4) | 5.012 (0.025)* |
| Absence | 198 (81.5) | 153 (77.3) | 45 (22.7) | | 107 (54.0) | 91 (46.0) | |
| **Academic Performance** | | | | | | | |
| Never failed an exam | 214 (88.1) | 158(73.8) | 56 (26.2) | 1.770 (0.183) | 104 (48.6) | 110 (51.6) | 2.925 (0.087) |
| Failed an exam | 29 (11.9) | 18 (62.1) | 11 (37.9) | | 19 (65.5) | 10 (34.5) | |
| **Academic Satisfaction** | | | | | | | |
| Satisfied | 155 (63.8) | 115 (74.2) | 40 (25.8) | 0.668 (0.414) | 84 (54.2) | 71 (45.8) | 2.190 (0.139) |
| Not satisfied | 88 (36.2) | 61 (69.3) | 27(30.7) | | 39 (44.3) | 49 (55.7) | |
| **Perceived Relationship with Parents** | | | | | | | |
| Good relationship | 192 (79.0) | 149 (77.6) | 43 (22.4) | 12.274 (<0.001)** | 103 (53.6) | 89 (46.4) | 3.357 (0.067) |
| Poor relationship | 51 (21.0) | 27 (52.7) | 24 (47.1) | | 20 (39.2) | 31 (60.8) | |
| **Perceived Relationship with Friends** | | | | | | | |

(*Continued*)

**Table 2.** (Continued)

| Variables | n (%) | Sleep Quality | | X² (p-value) | Internet Addiction | | X² (p-value) |
|---|---|---|---|---|---|---|---|
| | | Good n (%) | Poor n (%) | | Normal internet use n (%) | Internet addiction n (%) | |
| Good relationship | 144 (59.3) | 111 (77.1) | 33 (22.9) | 3.836 (0.050) | 86 (59.7) | 58 (40.3) | 11.722 (0.001)* |
| Poor relationship | 99 (40.7) | 65 (65.7) | 34 (34.3) | | 37 (37.4) | 62 (62.6) | |
| **Perceived Relationship with Teachers** | | | | | | | |
| Good relationship | 116 (47.7) | 96 (82.8) | 20 (17.2) | 11.862 (0.001)* | 77 (66.4) | 39 (33.6) | 22.06 (<0.001)** |
| Poor relationship | 127 (52.3) | 80 (63.0) | 47 (37.0) | | 46 (36.2) | 81 (63.8) | |

*statistically significant at p<0.05

**statistically significant at p<0.001.

significant association was found between internet addiction and sleep duration, sleep efficiency, and the use of sleep medication (Table 3).

The independent variables found to have a statistically significant relationship with sleep quality and internet addiction in bivariate analysis were included in the final model for multiple logistic regression analysis after conducting the VIF test. The VIF test among the independent variables illustrated that the highest reported VIF was 2.470, indicating that there was no issue of multicollinearity. In regards to Poor sleep quality, independent factors such as internet addiction, family economic status, stressful life events, conflict in family and perceived relation with parent and teacher were subjected in a regression model. The adolescents indicating internet addiction were 1.845 times (aOR: 1.845, 95% CI: 1.344–3.608) at higher odds of experiencing poor sleep quality as compared to normal users. Likewise, as compared to participants who perceived to have good relations with their teachers, those who had poor relationships were twice (aOR: 2.274, 95% CI: 1.149 to 4.497) more likely to have poor sleep quality. Similarly, the adolescents who had experienced family conflict were twice (aOR: 2.355, 95% CI: 1.040–5.329) more at odds of experiencing poor sleep quality in comparison to their counterparts (Table 4).

In regards to internet addiction, sleep-related attributes including sleep latency, subjective sleep quality, sleep disturbance, daytime dysfunction, and factors such as stressful life event, conflict in family and perceived relation with parent and teacher were subjected in a regression model. Bivariate analysis illustrated that attributes such as long sleep latency, bad or fairly bad subjective sleep quality, frequent sleep disturbance, and frequent daytime dysfunction could lead to higher odds of internet addiction. However, this association weakened after accounting for other factors in the adjusted model. The odds of internet addiction were found to be increased by five-fold (aOR: 5.613, 95% CI: 2.007–15.704) in adolescents with fairly bad or bad subjective sleep quality as compared to their counterparts. Similarly, those experiencing sleep disturbance were 1.781 times (aOR: 1.781, 95% CI: 1.251–4.872) more at odds of internet addiction. The adolescents who reported having poor relationships with their teachers were twice (aOR = 2.298, 95% CI: 1.233–4.285) more at odds of having internet addiction as compared to those with good relationships. Likewise, those who have been exposed to family conflict were 1.606 times (aOR: 1.606, 95% CI: 1.2002–3.675) more at odds of experiencing internet addiction when compared to their counterparts (Table 5).

**Table 3. Internet addiction status according to sleep quality and sleep pattern.**

| Characteristics | n (%) | Internet Addiction | | X² (p-value) |
|---|---|---|---|---|
| | | Normal internet use | Internet addiction | |
| | | n (%) | n (%) | |
| **Overall Sleep Quality** | | | | |
| Good | 176 (72.4) | 99 (56.3) | 77 (43.8) | 8.102 (0.004)* |
| Poor | 67 (27.6) | 24 (35.8) | 43 (64.2) | |
| **Sleep Latency** | | | | |
| ≤15 minutes | 112 (46.1) | 66 (58.9) | 46 (41.1) | 5.744 (0.047)* |
| 16–30 minutes | 87 (35.8) | 38 (43.7) | 49 (56.3) | |
| ≥30 minutes | 44 (18.1) | 19 (43.2) | 25 (56.8) | |
| **Subjective Sleep Quality** | | | | |
| Good | 63 (25.9) | 42 (66.7) | 21 (33.3) | 28.314 (<0.001)** |
| Fairly good | 142 (58.4) | 76 (53.5) | 66 (46.5) | |
| Fairly bad | 24 (9.9) | 3 (12.5) | 21 (87.5) | |
| Bad | 14 (5.8) | 2 (14.3) | 12 (85.7) | |
| **Sleep Duration** | | | | |
| ≥7 hours | 171 (70.4) | 87 (50.9) | 84 (49.1) | 0.174# (0.917) |
| 5–6 hours | 65 (26.7) | 33 (50.8) | 32 (49.2) | |
| <5 hours | 7 (2.9) | 3 (42.9) | 4 (57.1) | |
| **Sleep Efficiency** | | | | |
| ≥85% | 233 (95.9) | 120 (51.5) | 113 (48.5) | 1.841 (0.398) |
| 65–84% | 7 (2.9) | 2 (28.6) | 5 (71.4) | |
| <65% | 3 (1.2) | 1 (33.3) | 2 (66.7) | |
| **Sleep Disturbance** | | | | |
| Not during the past month | 28 (11.5) | 20 (71.4) | 8 (28.6) | 12.582(0.002)* |
| Less than once a week | 144 (59.3) | 78 (54.2) | 66 (45.8) | |
| More than once a week | 71 (29.2) | 25 (35.2) | 46 (64.8) | |
| **Use of Sleep Medication** | | | | |
| Not during the past month | 228 (93.8) | 116 (50.9) | 112 (49.1) | 8.610 (0.056) |
| Less than once a week | 10 (4.1) | 7 (70.0) | 3 (30.0) | |
| More than once a week | 5 (2.1) | 0 (0) | 5 (100.0) | |
| **Daytime Dysfunction** | | | | |
| Never | 86 (35.4) | 55 (64.0) | 31 (36.0) | 17.566 (0.001)* |
| Less than once a week | 109 (44.9) | 55 (50.5) | 54 (49.5) | |
| 1–2 times per week | 38 (15.6) | 11 (28.9) | 27 (71.1) | |
| ≥3 times per week | 10 (4.1) | 2 (20.0) | 8 (80.0) | |

*statistically significant at p<0.05

**statistically significant at p<0.001.

## Discussion

This study investigated the prevalence and risk factors of poor sleep quality and internet addiction among school-going adolescents in Dakshinkali Municipality, Nepal. Nearly one-fourth (27.6%) of the adolescents exhibited poor sleep quality, aligning with previous studies in Nepal where poor sleep quality among adolescents ranged from 24.4% to 39.1% [7,19,29]. This prevalence is slightly lower than past prevalence reported among higher-secondary and undergraduate students in Nepal, where the rate of poor sleep quality ranged from 31% to 59.1% [7,19–22]. Globally, the prevalence of poor sleep quality among adolescents has been observed to

**Table 4. Multivariable analysis for factors associated with poor sleep quality.**

| Variables | Poor Sleep Quality cOR (95% CI) | p-value | Poor Sleep Quality aOR (95% CI) | p-value |
|---|---|---|---|---|
| **Internet Addiction** | | | | |
| Normal internet use | Ref | | Ref | |
| Internet addiction | 2.304 (1.288–4.120) | 0.005* | 1.845 (1.344–3.608) | 0.037* |
| **Family Economic Status** | | | | |
| Fifth quintile | Ref | | Ref | |
| Fourth quintile | 1.318 (0.448–3.879) | 0.616 | 0.948 (0.297–3.022) | 0.928 |
| Third quintile | 3.401 (1.264–9.151) | 0.015* | 2.046 (0.683–6.134) | 0.201 |
| Second quintile | 3.478 (1.302–9.286) | 0.013* | 1.746 (0.563–5.421) | 0.335 |
| First quintile | 2.562 (0.926–7.094) | 0.070 | 1.352 (0.405–4.510) | 0.623 |
| **Stressful Life Events** | | | | |
| No | Ref | | Ref | |
| Yes | 2.006 (1.033–3.897) | 0.040* | 1.124 (0.513–2.465) | 0.770 |
| **Perceived Relationship with Parents** | | | | |
| Good | Ref | | Ref | |
| Poor | 3.080 (1.614–5.877) | 0.001* | 1.595 (0.739–3.443) | 0.234 |
| **Perceived Relationship with Teachers** | | | | |
| Good | Ref | | Ref | |
| Poor | 2.820 (1.545–5.146) | 0.001* | 2.274 (1.149–4.497) | 0.018* |
| **Conflict in Family** | | | | |
| No | Ref | | Ref | |
| Yes | 3.252 (1.660–6.371) | 0.001* | 2.355 (1.040–5.329) | 0.040* |

*statistically significant at p<0.05.

range from 24.0% to 53%, across different nations, irrespective of national economy [13,14,16,17]. The variation in the prevalence of poor sleep quality may be attributed to factors such as age, academic stress, access to technology, and study setting. All of these findings highlight the importance of addressing quality sleep among adolescents, as it can have significant consequences for academic performance, mental health, and overall quality of life.

In the context of internet addiction, nearly half (49.4%) of the adolescents scored above 30 on the Young's Internet Addiction Test (IAT), indicating potential internet addiction. This finding is consistent with a previous study from the year 2014 where 34.35% of Nepali adolescents were observed to have possible internet addiction under IAT cut-off ≥50 [30]. Similarly, another study from a peri-urban setting in Nepal reported around one-fifth (21.5%) of adolescents to be at the borderline of internet addiction while 13.3% had potential internet addiction [7]. Another study among undergraduate students reported a prevalence of internet addiction at 29.90% using the IAT cut-off score ≥50 [31]. The variations in the prevalence of internet addiction across studies underscore the challenges associated with comparing results as these discrepancies are mostly due to differences in operational definitions and cut-offs used for screening internet addiction. Various studies have employed different cut-off scores for the Internet Addiction Test (IAT) to define internet addiction. However, we chose to use the original cut-off provided for the IAT, as there was no evidence suggesting validations of alternative cut-offs for IAT in Nepali language. While the specific prevalence rates may vary, our findings, consistent with previous research, demonstrate a high prevalence of internet addiction and problematic internet use among Nepalese adolescents. This underscores the pressing need for developing and implementing effective interventions to address this growing concern.

**Table 5. Multivariable analysis for factors associated with internet addiction.**

| Variables | Internet Addiction cOR (95% CI) | p-value | Internet Addiction aOR (95% CI) | p-value |
|---|---|---|---|---|
| **Sleep Latency** | | | | |
| ≤15 minutes | **Ref** | | **Ref** | |
| >15 minutes | 1.863 (1.118–3.105) | 0.017* | 1.242 (0.681–2.268) | 0.480 |
| **Subjective Sleep Quality** | | | | |
| Good/ fairly good | Ref | | Ref | |
| Bad/ fairly bad | 8.952 (3.358–23.864) | <0.001** | 5.613 (2.007–15.701) | 0.001* |
| **Sleep Disturbance** | | | | |
| Not during the past month | Ref | | Ref | |
| During past month | 2.718 (1.148–6.440) | 0.023* | 1.781 (1.251–4.872) | 0.031* |
| **Daytime Dysfunction** | | | | |
| Never | Ref | | Ref | |
| Less than once a week | 1.742 (0.977–3.107) | 0.060 | 1.127 (0.576–2.208) | 0.727 |
| More than once per week | 4.777 (2.203–10.357) | <0.001** | 1.902 (1.083–4.638) | 0.015* |
| **Stressful Life Events** | | | | |
| No | Ref | | Ref | |
| Yes | 2.432 (1.253–4.720) | 0.009* | 1.451 (0.652–3.231) | 0.361 |
| **Perceived Relationship with Friends** | | | | |
| Good | Ref | | Ref | |
| Poor | 2.485 (1.468–4.204) | 0.001* | 1.436 (0.761–2.712) | 0.264 |
| **Perceived Relationship with Teachers** | | | | |
| Good | Ref | | Ref | |
| Poor | 3.477 (2.049–5.898) | <0.001** | 2.298 (1.233–4.285) | 0.009* |
| **Conflict in Family** | | | | |
| No | Ref | | Ref | |
| Yes | 2.131 (1.389–4.170) | 0.027* | 1.606 (1.202–3.675) | 0.026* |

*statistically significant at p<0.05

**statistically significant at p<0.001.

Sociodemographic factors such as participant's age, gender, and parental educational levels did not show a statistically significant association with adolescent sleep quality or internet addiction in this study. This aligns with previous research conducted in Nepal, where gender was not linked to either sleep quality or internet addiction [7,19,31]. Similarly, studies world-wide have reported that there are no gender-related differences in sleep quality or internet addiction [13,17]. Likewise, studies conducted among adolescents from different nations have also found no significant relationship between adolescent sleep quality or internet addiction and parental education levels [14,17,19]. However, family wealth was initially linked to poor sleep quality in bivariate analysis, which was weakened in the adjusted model. In contrast, the existing relationship between family wealth and sleep quality among adolescents has been highlighted by past studies [16,32]. This might be attributed to the fact that financial constraints may limit families' ability to prioritize sleep, increase worries and concerns among parents, and might give rise to stressful family environments which, in turn, can disrupt sleep patterns and contribute to sleep disturbances among adolescents [33,34]. However, other factors beyond economic status may also influence the relationship between family wealth and sleep quality, warranting further investigation.

The perceived relationship of the adolescents with their teachers emerged as a significant predictor for both poor sleep quality and internet addiction. This aligns with a previous study from Nepal where adolescents' perceived relationship with teachers was linked to sleep quality [19]. It has been observed that adolescents with poor relationships with their teachers are more likely to experience poor sleep quality compared to those with positive rapport with their teachers [16,35]. Similarly, poor teacher support has been observed to be an important predictor of internet addiction among adolescents [36]. These findings highlight the importance of school environment and positive teacher-student relationships in not only enhancing academic success but also influencing the psychological well-being of students, including their sleep quality and internet addiction. While this study did not delve into the impact of school environment on overall psychological well-being, it presents a potential area for future research by prospective researchers.

The presence of family conflict has been identified as an independent predictor for both poor sleep quality and internet addiction. Adolescents who have poor family relationships are more likely to have sleep disturbances [16,37]. Poor family functioning such as high levels of conflict and poor family support has been suggested to be leading factors for excessive internet use among adolescents [36]. It has also been observed that adolescents exposed to domestic violence are more likely to experience smartphone addiction [38]. The individuals exposed to conflict might resort to the internet as an alternative to avoid the situation and/or to forget dreadful events, alleviate feelings of loneliness, and cope with the fear of rejection, ultimately affecting their sleep cycle [39,40]. Therefore, addressing family environment is crucial for promoting better psychological well-being, as adolescents exposed to family conflicts are more prone to poor sleep quality and internet addiction.

A statistically significant relationship was observed between overall sleep quality and internet addiction among adolescents, consistent with previous research linking internet addiction to sleep quality [24,41]. Digital screen and internet addiction have been observed to have a significant relationship with poor sleep quality among adolescents [41]. A study from a peri-urban Nepal also revealed that around one-third of adolescents with borderline internet addiction experienced poor sleep quality [7]. Extending past research on the link between internet addiction and overall sleep quality, our study further aimed to explore the relationship of specific sleep attributes such as sleep latency, subjective sleep quality, duration, efficiency, disturbances, medication use, and daytime dysfunction with internet addiction. Notably, multivariate analysis identified significant associations between internet addiction and subjective sleep quality, sleep disturbances, and daytime dysfunction, even when controlling for other factors. Similar associations between poor subjective sleep quality and internet addiction have been noted in studies from various countries like Taiwan, Turkey, Japan, and Ethiopia [2,42–44]. These findings suggest that excessive internet use can disrupt overall sleep quality and lead to dissatisfaction with sleep experience.

It was observed that sleep disturbance has a statistically significant relation with internet addiction. This is in line with past studies where higher secondary and undergraduate students from Nepal reported a loss of sleep due to experience of internet addiction and late-night internet use [20,24]. Youngsters often surf the internet to cope with sleepless nights, leading to habitual night-time internet use, and increased screen time, which may contribute to sleep disturbances [18,45]. A study among Chinese adolescents suggested that excessive internet use is associated with increased sleep problems, such as difficulty falling asleep and maintaining sleep, and experiencing insomnia among individuals with internet addiction [46]. Internet addiction was found to have a significant relationship with daytime dysfunction among adolescents. This finding is consistent with previous research linking excessive internet use to cognitive impairments and reduced daytime functioning such as daytime sleepiness, impaired

concentration, and reduced productivity [2,47,48]. All of these findings suggest excessive use of the internet can disrupt the natural sleep-wake cycle and contribute to sleep problems, thus, it is important to recognize the potential consequences of internet addiction on daytime functioning and productivity, which can affect sleep quality in adolescents.

While this study is among the few studies that have assessed the relationship between sleep quality-related attributes and internet addiction among adolescents in the Nepali context, it is crucial to acknowledge its inherent limitations, and the findings should be interpreted accordingly. Despite our efforts to mitigate reporting biases in this self-reporting survey, through the provision of a private space at the school during data collection and conducting an orientation session to clarify each survey question, still there might be some biases introduced from factors such as social desirability, recall bias, or varying interpretations of symptoms in the survey questions. Future studies could administer more objective measures, to validate the self-reported data, which we failed to perform due to resource constrain. Additionally, the study did not examine whether individuals with poor sleep quality were taking daytime naps or how this might have impacted their academic performance. Furthermore, the study did not explore the potential impact of the Nepalese school environment, emotional status of the participants, physical heath, physical activity, daily sleep routine, screen time, academic load and cultural factors on sleep quality and internet addiction. Likewise, variations in internet usage and sleep quality across weekends or holidays, which might provide valuable behavioral insights, were not explored. These factors may vary across different cultures and societies, and which is an important area for future research to provide a more comprehensive understanding of the relationship between sleep quality, internet addiction, and cultural context. Additionally, the cross-sectional design limits our ability to establish causal relationships between sleep quality and internet addiction. Future research employing longitudinal designs and Structural Equation Modeling (SEM) could elucidate causal pathways between these variables.

## Conclusion

The study found a significant association between poor sleep quality and internet addiction among school-going adolescents in Dakshinkali Municipality, Nepal. A statistically significant relationship was observed between overall sleep quality and internet addiction levels, whereas specific sleep attributes such as subjective sleep quality, disturbances, and daytime impairments were strongly linked to internet addiction. The study also identified psychological risk factors like family conflicts and poor teacher relationships as important predictors of poor sleep quality and internet addiction. The findings highlight the growing public health concern of poor sleep quality and internet addiction among adolescents, which illustrate the need for targeted interventions focusing on digital wellness, positive teacher-student relationships, alongside shield adolescents from family conflicts, and implement practical approaches that could expedite mental health status and address the probable mental health issues among the adolescents within schools,

## Supporting information

**S1 Data.**
(XLSX)

## Acknowledgments

We thank all the study participants and their parents, and school authorities for their valuable time, without their support this study wouldn't have been possible.

## Author Contributions

**Conceptualization:** Sophiya Acharya, Anisha Chalise, Shishir Paudel.

**Data curation:** Sophiya Acharya, Anisha Chalise, Shishir Paudel.

**Formal analysis:** Sophiya Acharya, Anisha Chalise, Shishir Paudel.

**Funding acquisition:** Sophiya Acharya.

**Investigation:** Sophiya Acharya, Shishir Paudel.

**Methodology:** Sophiya Acharya, Anisha Chalise, Nirmal Raj Marasine, Shishir Paudel.

**Project administration:** Sophiya Acharya, Shishir Paudel.

**Resources:** Sophiya Acharya, Shishir Paudel.

**Supervision:** Anisha Chalise, Shishir Paudel.

**Validation:** Sophiya Acharya, Anisha Chalise, Nirmal Raj Marasine, Shishir Paudel.

**Visualization:** Sophiya Acharya, Anisha Chalise, Shishir Paudel.

**Writing – original draft:** Sophiya Acharya, Anisha Chalise, Shishir Paudel.

**Writing – review & editing:** Sophiya Acharya, Anisha Chalise, Nirmal Raj Marasine, Shishir Paudel.

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
