## [Decision Letter · Decision Letter 0]

28 May 2024

PONE-D-24-11194Sleep Quality and Internet Addiction among Adolescents in Dakshinkali Municipality, NepalPLOS ONE

Dear Dr. Paudel,

Thank you for submitting your manuscript to PLOS ONE. After careful consideration, we feel that it has merit but does not fully meet PLOS ONE’s publication criteria as it currently stands. Therefore, we invite you to submit a revised version of the manuscript that addresses the points raised during the review process.

We look forward to receiving your revised manuscript.

Kind regards,

Humayun Kabir

Academic Editor

PLOS ONE

Journal Requirements:

Reviewers' comments:

Reviewer's Responses to Questions

**Comments to the Author**

1. Is the manuscript technically sound, and do the data support the conclusions?

Reviewer #1: Partly

2. Has the statistical analysis been performed appropriately and rigorously? 

Reviewer #1: No

3. Have the authors made all data underlying the findings in their manuscript fully available?

Reviewer #1: Yes

4. Is the manuscript presented in an intelligible fashion and written in standard English?

Reviewer #1: Yes

5. Review Comments to the Author

Reviewer #1: The reviewer’s comments for the manuscript:

Title: The specific research objectives are not evident from the title. Please revise accordingly.

Methodology:

(1) The assessment tools should be detailed separately under the 'Measurements' section.

(2) Please provide a description of the forward and backward translation process to ensure the reliability and validity of the tools.

(3) The regression analysis section should clarify the dependent and independent variables.

(4) Adjust the analysis method in Tables 2 and 3 as follows: If the original data are continuous variables, use t-tests or ANOVA to analyze the differences between groups. Additionally, the authors do not clarify the cut-off points for each variable and their rationales, such as sleep quality, severity of internet addiction, and the quality of family and teacher-student relationships.

(5) The relationships between sleep and internet addiction may need to be explored using Structural Equation Modeling (SEM).

(6) Important variables related to adolescent sleep issues were omitted during data collection, including emotional factors (e.g., anxiety, depression), physical health, daily routines, academic load, and extracurricular activities. Thus, the explanatory power of the study results is limited. It is recommended to include these variables.

Overall Judgment: Recommend a Major Revision.

6. PLOS authors have the option to publish the peer review history of their article (what does this mean?). If published, this will include your full peer review and any attached files.

Reviewer #1: No

---

## [Author Response · Author response to Decision Letter 0]

6 Jun 2024

The reviewer’s comments for the manuscript:

Title: The specific research objectives are not evident from the title. Please revise accordingly.

Thank you for your valuable feedback. Considering your comment, we have revised our article title as “Exploring the Association between Sleep Quality, Internet Addiction, and Related Factors among Adolescents in Dakshinkali Municipality, Nepal” making it more reflective to our research objectives. 

Methodology:

(1) The assessment tools should be detailed separately under the 'Measurements' section.

Thank you for your suggestion regarding the assessment tools. We have reorganized the methodology section and created a separate "Measurements" sub-section. This section now provides detailed information about PSQI and IAT along with the cutoff we used to categorize the attributes.

(2) Please provide a description of the forward and backward translation process to ensure the reliability and validity of the tools.

Thank you for notifying this missing information. We have tried to add some details on process adopted for translation and back-translation of the tool. 

(3) The regression analysis section should clarify the dependent and independent variables.

Thank you for suggestion, we have tried to make it more clear in the regression analysis part within the result section. 

(4) Adjust the analysis method in Tables 2 and 3 as follows: If the original data are continuous variables, use t-tests or ANOVA to analyze the differences between groups. Additionally, the authors do not clarify the cut-off points for each variable and their rationales, such as sleep quality, severity of internet addiction, and the quality of family and teacher-student relationships.

Thank you for your feedback. We employed clear cutoffs to categorize sleep quality and internet addiction severity (which me have tried to clear out in methodology section.) This approach allows for better comparability with national and international studies that have adopted similar categorization methods. Additionally, using chi-square allows us to explore the differences in variables across these categories. 

In regards to cut-off points for sleep quality, and internet addiction, we used cutoff suggested by original tool and previous studies. 

Cutoff we used for Sleep quality 

o PSQI is a four-point Likert scale ranging from 0 to 3 for each 19 items with the overall PSQI score ranging from 0 to 21. The lower score represents better sleep quality and the cut-off value of ≤5 can be used to denote good sleep quality


https://www.sciencedirect.com/science/article/abs/pii/0165178189900474?via%3Dihub


https://www.sciencedirect.com/science/article/abs/pii/S138994570600551X?via%3Dihub

Cutoff we used for Internet Addiction 

o Scores obtained from the IAT are grouped into four categories: normal (0–30), mild IA (31–49), moderate IA (50–79), and severe IA (80–100).

• https://www.iitk.ac.in/counsel/resources/IATManual.pdf COPYRIGHT © by Dr. Kimberly S. Young All rights reserved. Center for Internet Addiction Recovery, P.O. Box 632, Bradford, PA 16701 

• Internet addiction validation study in Indonesian adolescents (https://www.ncbi.nlm.nih.gov/pmc/articles/PMC7861384/#:~:text=The%20total%20scores%20of%20IAT,)%20%5B5%2C19%5D). 

• Kumar N, Kumar A, Mahto SK, et al. Prevalence of excessive internet use and its correlation with associated psychopathology in 11th and 12th grade students. General Psychiatry 2019;32:e100001. doi:10.1136/ gpsych-2018-100001

Subjective sleep quality, Sleep duration, Sleep efficiency, Sleep disturbance, use of sleep medications and Day time dysfunction are derived from 19 self-reported items of PSQI using the suggested procedure 

o https://www.goodmedicine.org.uk/files/assessment,%20pittsburgh%20psqi.pdf (Buysse, DJ, Reynolds CF, Monk TH, Berman SR, Kupfer DJ: The Pittsburgh Sleep Quality Index (PSQI): A new instrument for psychiatric research and practice. Psychiatry Research 28:193-213, 1989) 

o We have added a sentence in measurement sub-section under Data collection section, to make it more clear for the readers. Thank you for mentioning this missing detail

In regards to Participants relation with their Friends, Family and Teachers, rather than looking at quality of family and teacher-student relationships we had simply asked participant to rate their Perceived nature of relation with them. We have tried to make it more clear in the manuscript Data collection section. 

(5) The relationships between sleep and internet addiction may need to be explored using Structural Equation Modeling (SEM).

Thank you for this suggestion, we highly appreciate it. Structural Equation Modeling (SEM) can be a powerful tool for examining complex relationships. However, in our cross-sectional study design, we are not looking to establish causal relationships. Our study focused on assessing associations between sleep quality and internet addiction so we went for Regression analysis. We agree that future research employing longitudinal designs and SEM to elucidate causal pathways between these variables would be beneficial. Considering your feedback, we have acknowledged this limitation in our manuscript.

(6) Important variables related to adolescent sleep issues were omitted during data collection, including emotional factors (e.g., anxiety, depression), physical health, daily routines, academic load, and extracurricular activities. Thus, the explanatory power of the study results is limited. It is recommended to include these variables.

Thank you for this valuable feedback. We acknowledge that including additional variables related to adolescent sleep issues, such as emotional factors (anxiety, depression), physical health, daily routines, academic load, and extracurricular activities, could provide a more comprehensive understanding of the factors influencing sleep quality among adolescents with internet addiction. We recognize the limitations of our current study and the potential benefits of including these variables in future research. We have incorporated this limitation in the revised manuscript.

---

## [Decision Letter · Decision Letter 1]

21 Nov 2024

PONE-D-24-11194R1Exploring the Association between Sleep Quality, Internet Addiction, and Related Factors among Adolescents in Dakshinkali Municipality, NepalPLOS ONE

Dear Dr. Paudel,

Thank you for submitting your manuscript to PLOS ONE. After careful consideration, we feel that it has merit but does not fully meet PLOS ONE’s publication criteria as it currently stands. Therefore, we invite you to submit a revised version of the manuscript that addresses the points raised during the review process.

We look forward to receiving your revised manuscript.

Kind regards,

Runtang Meng, PhD

Academic Editor

PLOS ONE

Journal Requirements:

Reviewers' comments:

Reviewer's Responses to Questions

**Comments to the Author**

1. If the authors have adequately addressed your comments raised in a previous round of review and you feel that this manuscript is now acceptable for publication, you may indicate that here to bypass the “Comments to the Author” section, enter your conflict of interest statement in the “Confidential to Editor” section, and submit your "Accept" recommendation.

Reviewer #2: All comments have been addressed

Reviewer #3: (No Response)

Reviewer #4: (No Response)

2. Is the manuscript technically sound, and do the data support the conclusions?

Reviewer #2: Yes

Reviewer #3: Yes

Reviewer #4: Yes

3. Has the statistical analysis been performed appropriately and rigorously? 

Reviewer #2: Yes

Reviewer #3: Yes

Reviewer #4: Yes

4. Have the authors made all data underlying the findings in their manuscript fully available?

Reviewer #2: Yes

Reviewer #3: Yes

Reviewer #4: Yes

5. Is the manuscript presented in an intelligible fashion and written in standard English?

Reviewer #2: Yes

Reviewer #3: Yes

Reviewer #4: Yes

6. Review Comments to the Author

Reviewer #2: The reference numbers in the text need to be reviewed as they are following after full stops. The fonts seem to be different as well.

Reviewer #3: sample size: the author needs to justify the choice of 30% non response rate

Authors can elaborate and define the independent and dependent variables in the statistical analysis section

Ethical consideration: The author should make clear that participant can withdraw from the study at any point of time without having any repercussions on them or on their performance in the academic institution they belong to.

Reviewer #4: In the current manuscript titled " Exploring the Association between Sleep Quality, Internet Addiction, and Related Factors among Adolescents in Dakshinkali Municipality, Nepal", Acharya et al have explored the prevalence and contributing factors of poor sleep quality and internet addiction among adolescents in a Nepali community of Dakshinkali Municipality. Gathering data from 243 school adolescents using scaling tests to measure sleep quality and internet addiction, the authors have reported that a quarter of adolescents in the study were reported to experience poor sleep quality and nearly half screened positive for potential internet addiction. While the manuscript offers a lot of information that will be useful to the sleep, psychological, immunology and behavioral communities, there are following comments that need to be addressed to further improve the quality of this article:

1. In the "Sampling Size and Sampling Procedure", please describe briefly what was the non-response rate of 30% due to and what could be done to further improve this rating in the future studies?

2. In "Results and Table 3", please describe briefly how often were the occurrences of sleep deprivation or poor sleep quality due to internet addiction (1-2 times a week, daily etc.?) Was this measured in the tests?

3. Were the poor sleep quality individuals taking any naps during the day and was this affecting their overall performance in the school exams?

4. In "Results and Table 4", was exercise or any physical activity (riding a bicycle to school) recorded among the participants and take into consideration with the survey results? Please describe briefly.

5. In "Results and Table 5", did the tests include questions that could inform if the poor sleep quality and internet usage among the adolescents in the study was higher during the weekends or holiday period compared to weekdays? Please describe briefly.

7. PLOS authors have the option to publish the peer review history of their article (what does this mean?). If published, this will include your full peer review and any attached files.

Reviewer #2: No

Reviewer #3: No

Reviewer #4: No

---

## [Author Response · Author response to Decision Letter 1]

22 Nov 2024

We would like to thank all the reviewers for their valuable time and comments provided for improvement of our manuscript. We hereby share with you our response to all the comments provided. Thank you 

Reviewer #2: 

The reference numbers in the text need to be reviewed as they are following after full stops. The fonts seem to be different as well.

Thank you for notifying us about the placement of references and font consistency. We have revised the manuscript to place all reference numbers before the full stops. Additionally, we have ensured uniformity in font usage across the manuscript, adhering to Times New Roman throughout.

Reviewer #3:

 sample size: the author needs to justify the choice of 30% non-response rate

Authors can elaborate and define the independent and dependent variables in the statistical analysis section

Thank you for your valuable time and suggestions. In regards to 30% non-response rate, it is the part of our initial methodology, we used the maximum limit of non-response rate that is usually practiced to optimize the sample size. We wanted keep our sample as optimum as possible as it was unclear how many participants will be able to provide their complete response in self-administered questionnaire. As mentioned in our result section our achieved response rate is 87% as among 280 participants approached, 243 provided complete response (yielding a non-response rate of 13%). This way we were able to secure the minimum required sample of 215.

Thank you for this suggestion. We have revised the measurements section to explicitly define the independent variables and dependent variables.

Ethical consideration: The author should make clear that participant can withdraw from the study at any point of time without having any repercussions on them or on their performance in the academic institution they belong to.

Thank you for your valuable suggestion. The right to withdraw at any point in the data collection was respected, which is the reason for getting complete response from 243 while 280 were approached. We have added the missing information about right to withdraw in ethical statement section. Thank you. 

Reviewer #4:

In the current manuscript titled "Exploring the Association between Sleep Quality, Internet Addiction, and Related Factors among Adolescents in Dakshinkali Municipality, Nepal", Acharya et al have explored the prevalence and contributing factors of poor sleep quality and internet addiction among adolescents in a Nepali community of Dakshinkali Municipality. Gathering data from 243 school adolescents using scaling tests to measure sleep quality and internet addiction, the authors have reported that a quarter of adolescents in the study were reported to experience poor sleep quality and nearly half screened positive for potential internet addiction. While the manuscript offers a lot of information that will be useful to the sleep, psychological, immunology and behavioral communities, there are following comments that need to be addressed to further improve the quality of this article:

Thank you for your valuable time and feedback provided to improve our manuscript. 

1. In the "Sampling Size and Sampling Procedure", please describe briefly what was the non-response rate of 30% due to and what could be done to further improve this rating in the future studies

The right to withdraw at any point in the data collection was respected which is the reason for getting complete response from 87% of approached participants. The non-response of 30% was the part of our methodology section as we were unsure how many participants will provide complete response and how many participants under 17 years will bring signed parental consent. So, to be on the safe side and meet the minimum sample of 215, we adjusted 30% non-response during the calculation of our sample size. However, the true non-response during the data collection of this study was 13%. We feel that we need to respect the voluntary participation of the participants, and at the same time not meeting the minimum required sample could lead to misrepresentation of the findings, so we adjusted the non-response to optimize the sample so that minimum sample could be secured. If there are no time constrains, then non-response rates can also be minimized by incorporating enhanced participant engagement strategies, such as sending reminders, follow up with the participants reporting missing information and parental involvement workshops, before the data collection process.

2. In "Results and Table 3", please describe briefly how often were the occurrences of sleep deprivation or poor sleep quality due to internet addiction (1-2 times a week, daily etc.?) Was this measured in the tests?

Thank you for your comment. In this regards the sleep deprivation and poor sleep quality were assessed as a part of Pittsburgh Sleep Quality Index, while the internet addiction was measured through Young’s Internet Addiction Test. The value of poor sleep quality vs. internet addition is based on cross-tabulation. Due to being a cross-sectional study, we cannot specify how often the occurrences of sleep deprivation is due to internet addiction, and we can only reflect what is the distribution of different attributes of sleep deprivation across internet addition. 

3. Were the poor sleep quality individuals taking any naps during the day and was this affecting their overall performance in the school exams?

Thank you. While daytime dysfunction was assessed, specific data regarding daytime napping and its impact on academic performance were not collected. We have added this as a recommendation for future research in the limitations section.

4. In "Results and Table 4", was exercise or any physical activity (riding a bicycle to school) recorded among the participants and take into consideration with the survey results? Please describe briefly.

No, we have not measured the attributes related to physical activity in our study. Our study area Dakshinkali Municipality lies in the mountainous regions where taking bicycle to school is not common as compared to Terai regions of Nepal, so it never occurred to us. In regards to other physical activities such as school games, daily exercise and other, initially we had planned to explore these factors but considering the volume and nature of questionnaire (self-administered) and complexity to properly report physical activity in terms of their frequency, intensity, duration and diverse nature of activity, we did not assess it as we were more focused on internet addition and sleep. We have addressed this in the limitation section. Thank you. 

5. In "Results and Table 5", did the tests include questions that could inform if the poor sleep quality and internet usage among the adolescents in the study was higher during the weekends or holiday period compared to weekdays? Please describe briefly.

No, we did not specifically explore variations in internet usage and sleep quality across weekends or holidays. Recognizing the importance of these factors, we have recommended their inclusion in future research. Thank you.

---

## [Decision Letter · Decision Letter 2]

5 Jan 2025

Exploring the Association between Sleep Quality, Internet Addiction, and Related Factors among Adolescents in Dakshinkali Municipality, Nepal

PONE-D-24-11194R2

Dear Dr. Paudel,

We’re pleased to inform you that your manuscript has been judged scientifically suitable for publication and will be formally accepted for publication once it meets all outstanding technical requirements.

Kind regards,

Runtang Meng, PhD

Academic Editor

PLOS ONE

Additional Editor Comments (optional):

Reviewers' comments:

Reviewer's Responses to Questions

**Comments to the Author**

1. If the authors have adequately addressed your comments raised in a previous round of review and you feel that this manuscript is now acceptable for publication, you may indicate that here to bypass the “Comments to the Author” section, enter your conflict of interest statement in the “Confidential to Editor” section, and submit your "Accept" recommendation.

Reviewer #2: All comments have been addressed

Reviewer #4: All comments have been addressed

2. Is the manuscript technically sound, and do the data support the conclusions?

Reviewer #2: Yes

Reviewer #4: Yes

3. Has the statistical analysis been performed appropriately and rigorously? 

Reviewer #2: Yes

Reviewer #4: Yes

4. Have the authors made all data underlying the findings in their manuscript fully available?

Reviewer #2: Yes

Reviewer #4: Yes

5. Is the manuscript presented in an intelligible fashion and written in standard English?

Reviewer #2: Yes

Reviewer #4: Yes

6. Review Comments to the Author

Reviewer #2: All comments have been addressed that were asked by the reviewers. There is nothing to add to the current comments.

Reviewer #4: Thank you addressing all comments from the review that was done previously and updating the manuscript accordingly.

7. PLOS authors have the option to publish the peer review history of their article (what does this mean?). If published, this will include your full peer review and any attached files.

Reviewer #2: No

Reviewer #4: No

---

## [Editor Report · Acceptance letter]

8 Jan 2025

PONE-D-24-11194R2 

PLOS ONE

Dear Dr. Paudel, 

I'm pleased to inform you that your manuscript has been deemed suitable for publication in PLOS ONE. Congratulations! Your manuscript is now being handed over to our production team.

Kind regards, 

on behalf of

Dr. Runtang Meng 

Academic Editor

PLOS ONE